# The relationship between family history of cancer and cancer attitudes & beliefs within the Community Initiative Towards Improving Equity and Health Status (CITIES) cohort

Li Lin[1], Xiaochen Zhang[2], Mengda Yu[2], Brittany Bernardo[2], Toyin Adeyanju[2], Electra D. Paskett[1,2]*

1 College of Medicine, The Ohio State University, Columbus, Ohio, United States of America,
2 Comprehensive Cancer Center, The Ohio State University, Columbus, Ohio, United States of America

* electra.paskett@osumc.edu

## Abstract

### Objective

To determine the relationship between family history of cancer with cancer attitudes and beliefs (CABs) and cancer screening knowledge.

### Methods

This study used data collected for the Community Initiative Towards Improving Equity and Health Status (CITIES) project which surveyed Ohioans ages 21–74. In the current analysis, we included data on age, gender, race, marital status, education, income, financial security, health insurance, CABs, knowledge about the correct age to begin cancer screenings, and presence of a first-degree relative with cancer. Multivariable logistic regression was used to examine the association of family history of cancer with CABs and knowledge about the correct age to begin cancer screening.

### Results

Participants were predominantly over the age of 41, female, and white. Out of 603 participants, 295 (48.92%) reported not having a first-degree relative with cancer and 308 (51.08%) reported having a first-degree relative with cancer. Overall, 109 (18.08%) participants reported negative CABs, 378 (62.69%) reported moderate CABs, and 116 (19.24%) reported positive CABs. Participants who reported a first-degree relative with cancer were more likely to report positive CABs, but the association was not significant (p = .11). We observed that older, more educated, and married participants were more likely to have positive CABs (all p < 0.05). Family history of cancer was not associated with differences in knowledge about the correct age for beginning colorectal cancer screening (p = .85) and mammography (p = .88).

**Data Availability Statement:** The consent form for the CITIES project only allows for the reporting of the results as a group. De-identified data will be

available to access upon request with a proposal reviewed and approved by Cecilia DeGraffinreid (Cecilia.DeGraffinreid@osumc.edu) and a determination of exemption or approval by an Institutional Review Board (IRB). Upon receipt of the approvals, a Data Use Agreement will be completed. This form is a request for the specific data that is requested as well as a notice of all policies and rules about using the Community Initiative Towards Improving Equity and Health Status (CITIES) data.

**Funding:** This project was supported by The Ohio State University Comprehensive Cancer Center using Pelotonia funds and a supplement to the National Cancer Institute Grant P30 CA016058, The Ohio State University Center for Clinical and Translational Science (funded by the National Center for Advancing Translational Sciences of the National Institutes of Health under Grant UL1TR00273), and the Recruitment, Intervention and Survey Shared Resource (RISSR) at The Ohio State University Comprehensive Cancer Center (funded by the National Cancer Institute Grant P30 CA016058). This work was also supported by the Samuel J. Roessler Memorial Scholarship through The Ohio State University College of Medicine's Medical Student Research Program Scholarship (To LL) and the National Cancer Institute (F99CA25374501 to XZ). There was no additional external funding received for this study. The funders had no role in study design, data collection and analysis, decision to publish, or reparation of the manuscript.

**Competing interests:** The authors have declared no competing interests exist.

## Conclusions

Having a first-degree relative with cancer was not found to be associated with CABs or knowledge about cancer screening. However, age and socioeconomic status were associated with more positive CABs and increased knowledge about cancer screening. Future research should focus on standardizing a CABs scale and expanding the generalizability of our findings.

## Introduction

Cancer health disparities are influenced by a combination of factors including social conditions and policies, institutional context, social context, social relationships, physical context, individual demographics, individual risk factors, biological responses, and biological/genetic pathways [1]. These population-level characteristics can be collectively referred to as Social Determinants of Health and are an important target of public health interventions. Of the various models that have attempted to elucidate the mechanisms through which Social Determinants of Health occur, the Health Belief Model (HBM) is one of the most widely used frameworks.

The Health Belief Model (HBM) identifies beliefs that predict an individual's likelihood of engaging in health preventive behaviors [2–4]. Belief constructs of the HBM including perceived susceptibility, perceived severity, perceived benefits, and perceived barriers have been found to predict mammography and CRC screening adherence [4–6]. Related constructs such as negative attitudes and beliefs about cancer (e.g., cancer is automatically associated with death, not much can be done to prevent cancer) have been linked to increased rates of cancer incidence and mortality, lower economic status, and lower education level [7,8]. These fatalistic beliefs along with poor health literacy contribute to decreased engagement in preventative cancer care such as eating a healthy diet, using sun protection, avoiding tobacco, and following cancer screening guidelines [9–11]. Furthermore, negative emotions and attitudes are linked to an array of health consequences including anxiety, poor adherence to treatment, use of drugs and alcohol, disordered eating, and poor exercise [12–14].

Health beliefs are influenced and modified by factors such as age, gender, ethnicity, personality, socioeconomics, and knowledge [4]. However, studies about how cancer beliefs and behaviors vary in individuals who have a family member with cancer have yielded variable results. Some studies have found that while those who have a family member with cancer are more likely to participate in cancer screening, they are not any more likely to engage in other preventative behaviors such as smoking cessation, increasing physical activity, or changing their diet [15,16]. Intergenerational behavioral factors may be responsible for this finding. For instance, individuals with parents who smoke or eat unhealthy diets may be more likely to adopt these same behaviors [15].

Other evidence suggests that those with family history of cancer are more knowledgeable about screening guidelines, but their engagement with screening may lead them to believe that they no longer need to adopt lifestyle modifications [17]. Padamsee et al. (2020) found that the type of experience a person has with family history of cancer may affect their attitudes and behaviors towards cancer [18]. For example, women with close and traumatic experiences of cancer were more likely to engage in aggressive preventive actions, including surgery and chemoprevention, despite not having any known genetic mutations [18]. The link between family history of cancer, cancer knowledge, and cancer attitudes and beliefs remains unclear, but it is necessary for people to accurately understand their cancer risk and the important role that lifestyle factors play in reducing risk.

The purpose of the current study was to determine whether having a family history of cancer is associated with beliefs and attitudes about cancer and knowledge about cancer screening. We hypothesized that participants who reported a first-degree relative (FDR) with cancer had more negative beliefs and attitudes about cancer. In addition, we expected that family history of cancer would influence knowledge about cancer screening. Findings from this study would suggest a need for cancer prevention strategies that are informative about familial risk and increased education about the benefits of other modifiable cancer risk factors such as lifestyle changes.

## Methods

Data were collected from the Community Initiative Towards Improving Equity and Health Status (CITIES) project, part of an initiative by the National Cancer Institute (NCI) to assess population health in 15 NCI-designated Cancer Center catchment areas [19]. The CITIES project surveyed Ohio residents within the catchment area of The Ohio State University Comprehensive Cancer Center (OSUCCC), the state of Ohio. Sampling and data collection for this project have been detailed in previous studies and is summarized below [19,20]. Depending on the method of survey administration, participants provided verbal or written informed consent. This project was approved by the OSU Institutional Review Board in February 2017.

### Sampling and data collection

All Ohio residents aged 21 to 74 years were eligible for this study. Recruitment targeted underrepresented populations to ensure there were substantive percentages of racial/ethnic minorities, rural residents, and Appalachian Ohio residents. Survey administration occurred between May 30, 2017 to February 16, 2018. To recruit a diverse and representative group of participants, several methods were used including random selection from a Marketing Systems Group (white pages, commercial and United States Postal Service lists) and collaboration with community partners and events. Data was collected through phone calls, in-person interviews, and web surveys with translation used, as needed. Telephone interview respondents received an introductory letter about the study, followed by a telephone call from a trained interviewer one week later. Potential in-person interview respondents were approached individually or in a group setting where the study was explained. Informed consent was obtained either verbally for phone and in-person interviews or electronically for web surveys. Survey data was collected and managed with REDCap (Research Electronic Data Capture), a secure web-based data collection and survey system hosted by The Ohio State University [21].

### Family history of cancer

Family history of cancer was determined by a set of yes/no questions about first-degree family members: "Has your father ever been diagnosed with cancer?"; "Has your mother ever been diagnosed with cancer?"; "Do you have any brothers?", Have any been diagnosed with cancer?"; "Do you have any sisters?", Have any been diagnosed with cancer?"; Do you have any sons?", Have any been diagnosed with cancer?"; Do you have any daughters?", Have any been diagnosed with cancer?". Respondents were split into two groups: those with at least one first-degree relative with cancer and those with no first-degree relative with cancer.

### Outcomes: Cancer attitudes and beliefs

Cancer attitudes and beliefs (CABs) were assessed with five statements: (1) "It seems like everything causes cancer; (2) there's not much you can do to lower your chances of getting cancer;

(3) there are so many different recommendations about preventing cancer, it's hard to know which ones to follow; (4) when I think about cancer, I automatically think about death; and (5) cancer is most often caused by a person's behavior or lifestyle." Participants ranked the statements on a Likert scale: "1 = strongly agree, 2 = somewhat agree, 3 = somewhat disagree, 4 = strongly disagree." Answers to the five statements were then combined into a composite score using the method described by Vanderpool et al., 2019 [8]. Composite scores range from 5–20 with a lower score indicating more negative CABs and a higher score indicating more positive CABs. The last statement "Cancer is most often caused by a person's behavior or lifestyle," was reverse coded.

## Outcomes: Cancer knowledge

Knowledge about cancer screening practices was assessed with two open response questions: "At what age are most women supposed to start having mammograms?" and "At what age are most people supposed to start doing home blood stool tests, having a sigmoidoscopy or having a colonoscopy?" Participants were asked to respond with a numerical age. Participants who answered "50" were recoded as correct responses, and participants who answered any other numerical value were recoded as incorrect responses.

## Analysis

Demographic characteristics were summarized using frequencies for the categorical variables for all participants, and by FDR cancer status. Participant characteristics were compared between groups by FDR with cancer status using Chi-square tests for the categorical variables. Multivariable logistic regression models were used to examine the associations between FDR status with all the variables. To find significant covariates, we involved variables in a logistic regression model and backward selection method. An alpha level of removal of .10 was used. All analysis was done with a significance level of 0.05 and using SAS 9.4 [22].

## Results

Table 1 shows a demographic summary of the sample population (n = 603) for participants with and without a FDR with cancer. Overall, 32.01% were ages 51–65 years, 24.05% were ages 21–40 years, 22.55% were ages 41–50 years, and 21.39% were ages 66–74 years. More than half the sample was female (63.02%), predominantly white (65.17%), and the majority were married or living as married (66.33%). Additionally, 45.27% of participants were college graduates, 35.82% obtained a high school education or less, and 18.91% had technical school or some college. For household income, 39.47% of the sample earned $75,000 or more, 31.84% earned $35–74,999, and 28.69% earned less than $35,000. Related to financial security on present income, 42.12% indicated that they were living comfortably, 37.98% reported that they were getting by, and 19.9% indicated that they found it difficult. Most of the sample had health insurance (59.2% private, 30.85% public) and 9.95% were uninsured. For the CABs score, 62.69% scored between 11–15, 19.24% scored between 16–20, and 18.08% scored between 5–10.

Compared to participants without a FDR with cancer, participants with a FDR with cancer were older (p < 0.001), more likely to be white (p < 0.001), less likely to be single (p = .03), more likely to make $75,000 or more (p = .01), and more likely to have health insurance (p < 0.001). For CABs, participants without a FDR with cancer were more likely to score between 5–10 (22.37% vs. 13.96%) and less likely to score between 16–20 (16.27% vs. 22.08%) as compared to participants with a FDR with cancer (p = .01).

**Table 1. Demographic summary of participants with and without a first degree relative (FDR) with cancer.**

| Variable | No FDR With Cancer (n = 295) | Have FDR With Cancer (n = 308) | Total (n = 603) | P-value |
|---|---|---|---|---|
| **Age** | | | | |
| 21–40 years | 113 (38.31%) | 32 (10.39%) | 145 (24.05%) | < .001 |
| 41–50 years | 78 (26.44%) | 58 (18.83%) | 136 (22.55%) | |
| 51–65 years | 65 (22.03%) | 128 (41.56%) | 193 (32.01%) | |
| 66–74 years | 39 (13.22%) | 90 (29.22%) | 129 (21.39%) | |
| **Gender** | | | | |
| Male | 114 (38.64%) | 109 (35.39%) | 223 (36.98%) | .41 |
| Female | 181 (61.36%) | 199 (64.61%) | 380 (63.02%) | |
| **Race** | | | | |
| Hispanic | 43 (14.58%) | 18 (5.84%) | 61 (10.12%) | < .001 |
| Somali | 16 (5.42%) | 0 (0%) | 16 (2.65%) | |
| Asian | 17 (5.76%) | 7 (2.27%) | 24 (3.98%) | |
| African American | 56 (18.98%) | 53 (17.21%) | 109 (18.08%) | |
| White | 163 (55.25%) | 230 (74.68%) | 393 (65.17%) | |
| **Marital status** | | | | |
| Married/living as married | 198 (67.12%) | 202 (65.58%) | 400 (66.33%) | .03 |
| Divorced/widowed/separated | 46 (15.59%) | 70 (22.73%) | 116 (19.24%) | |
| Single/never married | 51 (17.29%) | 36 (11.69%) | 87 (14.43%) | |
| **Education** | | | | |
| High school or less | 108 (36.61%) | 108 (35.06%) | 216 (35.82%) | .61 |
| Tech school/some college | 51 (17.29%) | 63 (20.45%) | 114 (18.91%) | |
| College grad or higher | 136 (46.1%) | 137 (44.48%) | 273 (45.27%) | |
| **Income** | | | | |
| <$35,000k | 101 (34.24%) | 72 (23.38%) | 173 (28.69%) | .01 |
| $35,000–74,999 | 86 (29.15%) | 106 (34.42%) | 192 (31.84%) | |
| ≥$75,000 | 108 (36.61%) | 130 (42.21%) | 238 (39.47%) | |
| **Financial security** | | | | |
| Finding it difficult on present income | 71 (24.07%) | 49 (15.91%) | 120 (19.9%) | < .001 |
| Getting by on present income | 122 (41.36%) | 107 (34.74%) | 229 (37.98%) | |
| Living comfortably on present income | 102 (34.58%) | 152 (49.35%) | 254 (42.12%) | |
| **Health insurance** | | | | |
| None | 51 (17.29%) | 9 (2.92%) | 60 (9.95%) | < .001 |
| Private | 168 (56.95%) | 189 (61.36%) | 357 (59.2%) | |
| Public | 76 (25.76%) | 110 (35.71%) | 186 (30.85%) | |
| **Cancer attitudes and beliefs score** | | | | |
| 5–10 | 66 (22.37%) | 43 (13.96%) | 109 (18.08%) | .01 |
| 11–15 | 181 (61.36%) | 197 (63.96%) | 378 (62.69%) | |
| 16–20 | 48 (16.27%) | 68 (22.08%) | 116 (19.24%) | |

n = 603. Demographic characteristics were compared between groups with and without a FDR with cancer using Chi-square test for the categorical variables.

Table 2 presents the backward model selection used to examine the association between CABs with other variables. As compared to participants ages 21–40, participants ages 51–65 and ages 66–74 were respectively 1.74 times and 2.31 times more likely to have positive CABs (OR = 1.74, 95% CI = (1.07–2.81), p = 0.04; OR = 2.31, 95% CI = (1.34–3.97), p = 0.01). Compared to married or living as married participants, divorced, widowed, or separated and single or never married participants were respectively 48% and 58% less likely to have positive CABs (OR = 0.52, 95% CI (0.33–0.81), p = 0.35; OR = 0.42, 95% CI = (0.26–0.69), p = 0.04). As

**Table 2. The association of family history of cancer and demographic characteristics on cancer attitudes and beliefs.**

| Covariate | Odds Ratio | 95% Confidence Limits | | P-value |
|---|---|---|---|---|
| **Family history of cancer** | | | | |
| No FDR with cancer | ref | | | |
| Have FDR with cancer | 1.34 | .94 | 1.90 | .11 |
| **Age** | | | | |
| 21–40 years | ref | | | |
| 41–50 years | 1.40 | .86 | 2.28 | .53 |
| 51–65 years | 1.74 | 1.07 | 2.81 | .04 |
| 66–74 years | 2.31 | 1.34 | 3.97 | .01 |
| **Gender** | | | | |
| Female | ref | | | |
| Male | .74 | .52 | 1.05 | .09 |
| **Marital status** | | | | |
| Married/living as married | ref | | | |
| Divorced/widowed/separated | .52 | .33 | .81 | .35 |
| Single/never married | .42 | .26 | .69 | .04 |
| **Education** | | | | |
| High school or less | ref | | | |
| Tech school/some college | 1.89 | 1.18 | 3.02 | .58 |
| College grad or higher | 2.81 | 1.92 | 4.11 | < .001 |

n = 603. Backward model selection with an alpha level of removal of .10 was used. The following variables were removed from the model: Race, income, financial security, and insurance status.

compared to participants with a high school education or less, college grads were 2.81 times more likely to have positive CABs (OR = 2.81, 95% CI = (1.92–4.11), p < 0.001).

Table 3 presents the backward model selection of participants' knowledge about the correct age to begin CRC screening. Participants ages 51–65 were 4.55 times more likely to know the

**Table 3. The association of family history of cancer and demographic characteristics on knowledge of correct age to begin colorectal cancer screening.**

| Covariate | Odds Ratio | 95% Confidence Limits | | P-value |
|---|---|---|---|---|
| **Family history of cancer** | | | | |
| No FDR with cancer | ref | | | |
| Have FDR with cancer | .96 | .64 | 1.44 | .85 |
| **Age** | | | | |
| 21–40 years | ref | | | |
| 41–50 years | 2.56 | 1.49 | 4.40 | .46 |
| 51–65 years | 4.55 | 2.64 | 7.85 | < .001 |
| 66–74 years | 2.24 | 1.28 | 3.93 | .96 |
| **Income** | | | | |
| <$35,000 | ref | | | |
| $35,000–74,999 | 1.88 | 1.16 | 3.06 | .17 |
| ≥$75,000 | 2.81 | 1.26 | 3.29 | .05 |

n = 603. Backward model selection with an alpha level of removal of .10 was used. The following variables were removed from the model: Gender, race, marital status, education, financial security, and insurance status.

**Table 4. The association of family history of cancer and demographic characteristics on knowledge of correct age to begin mammography (female participants only).**

| Covariate | Odds Ratio | 95% Confidence Limits | | P-value |
|---|---|---|---|---|
| **Family history of cancer** | | | | |
| No FDR with cancer | ref | | | |
| Have FDR with cancer | .95 | .48 | 1.88 | .88 |
| **Age** | | | | |
| 21–40 years | ref | | | |
| 41–50 years | .94 | .26 | 3.39 | .06 |
| 51–65 years | 3.67 | 1.27 | 10.60 | .01 |
| 66–74 years | 3.69 | 1.12 | 11.45 | .02 |

n = 380. Backward model selection with an alpha level of removal of .10 was used. The following variables were removed from the model: Gender, race, marital status, income, financial security, education, insurance status.

correct age to begin CRC screening when compared to those ages 21–40 (OR = 4.55, 95% CI (2.64–7.85), p < 0.001). When compared to participants earning \$<35k annual income, those earning \$75k+ were 2.81 more likely to know the correct age to begin CRC screening (OR = 2.81, 95% CI = (1.26–3.29), p = 0.05).

Table 4 presents the backward model selection of female participants' knowledge about the correct age to begin mammography. As compared to participants ages 21–40 years of age, participants ages 51–65 and ages 66–74 were respectively 3.67 times and 3.69 times more likely to know the correct age to begin mammography (OR = 3.67, 95% CI (1.27–10.62), p = 0.01; OR = 3.69, 95% CI (1.12–11.45), p = 0.02).

## Discussion

In this study, we first examined the relationship between having a family history of cancer and CABs. We hypothesized that participants with a FDR with cancer would have more negative attitudes and beliefs about cancer. Out of 603 participants, about half (n = 308) reported having a FDR with cancer. Contrary to our hypothesis, more participants without a FDR with cancer scored negatively (5–10) on the CABs scale, and more participants with a FDR with cancer scored positively (16–20) on the CABs scale. However, participants with a FDR with cancer were not significantly more likely to have positive CABs. These findings are in contrast with a previous study which found that women with a FDR with breast cancer had more negative attitudes about breast cancer [23]. The difference in results could be due to the wide variety of experiences among participants with a FDR with cancer that were not taken into account. A previous study on women with a family history of breast or ovarian cancer found that the nature of their experience with family history of cancer affected cancer attitudes and health decisions [18]. Another factor to consider is that participants without a FDR with cancer may still have close experiences with cancer in other relatives or close friends [24,25].

In our analysis of the demographic variables of CABs, we found that older, more educated, and married participants were more likely to have positive CABs while younger, less educated, and single participants were less likely to have positive CABs. These findings are consistent with the HBM which identifies demographic variables affecting health beliefs and attitudes which in turn affect health behaviors and health outcomes [3,25]. Previous research has shown that single people have worse overall health outcomes and higher mortality as compared to married people, possibly due to differences in social support [26,27]. Protective effects of marriage in terms of improved survival rates have also been observed in certain cancers [28–30].

These findings suggest that marital status may affect health through effects on attitudes and beliefs, consistent with the HBM [2,3,31]. Older and more educated adults may also be more likely to have positive CABs due to increased knowledge about cancer screening and prevention.

However, it remains unclear whether attitudes about cancer translate into increased uptake of cancer prevention strategies. Studies have shown that women with a FDR with breast cancer experienced increased worry, increased perception of susceptibility, increased fear of breast cancer, and decreased perception of mammography barriers [23,32–34]. Negative CABs translated to increased uptake of cancer screening in some studies [23,35] while other studies did not find a difference in cancer screening uptake based on family history, regardless of perceived susceptibility and attitudes [32,33,36,37]. Another study found that family history of cancer was associated with increased cancer screening uptake but not other health promotive behaviors [38]. In general, all groups underestimated their breast cancer risk in relation to actual calculated risk, and engagement with cancer prevention was suboptimal [23,33,34]. Similar underestimation of cancer risk was found in studies of individuals with family history of colorectal cancer [39].

We also predicted that knowledge about cancer screening would differ between those with and without a FDR with cancer. Contrary to our hypothesis, cancer screening knowledge was unrelated to FDR cancer status. Rather, we found that older participants ages 51–65 years of age and participants earning more than $75,000 per year were more likely to know the correct age for beginning CRC screening. Older female participants were also more likely than younger female participants to know the correct age to begin mammography. A similar study found that older individuals were more likely to engage in early detection screening for breast cancer as compared to younger individuals, regardless of family history status [40]. These findings about breast and colorectal cancer knowledge may be due to current screening guidelines and the fact that older adults receive more targeted recommendations about breast and CRC screening [41]. Further research could examine cancer knowledge in younger adults by asking questions about Pap tests or skin cancer prevention which are health actions that are recommended beginning at a younger age [42]. Several studies have found that educational interventions based on the HBM can improve health knowledge and uptake of preventive health behaviors—including cancer screening [43–45].

## Strengths and limitations

Strengths of this study include a diverse set of participants although data was limited to just Ohioans. Questions and measures in this study have been used previously, allowing comparisons with the greater U.S. population through larger data sets. Another strength of this study is the use of logistic regression models to control for secondary variables.

The generalizability of our findings may be limited by geographic limitations and study design. Since the data was procured from a one-time cross-sectional survey, no formal sample size calculation was conducted post hoc. There is also bias introduced by the multiple data collection methods used. Therefore, characteristics of the study sample may differ from the general Ohio population in various ways.

In addition, the scoring method used for CABs has only been used in one previous study [8]. Each question of the measure is weighted equally with this method, but it is possible that some questions more accurately measure CABs than others. Our findings also suggest that cancer knowledge may be much more closely related to CABs than previously described.

In the future, it may be beneficial to look at each question of the CABs measure individually. For instance, the statement "When I think about cancer, I automatically think about

death" is closely tied to the construct of fatalistic health beliefs [9]. However, the statement "Cancer is most often caused by a person's behavior or lifestyle" may be more accurately measuring whether participants are knowledgeable about the fact that many cancers are preventable [46]. In future studies, it will be important to continue assessing the reliability and validity of the CABs questions.

## Conclusion

Overall, we did not find a difference in CABs or cancer knowledge between groups with and without a FDR relative with cancer. However, CABs and cancer knowledge did vary by other demographic variables that have been previously described in the HBM. Future areas of research should focus on assessing the reliability and validity of the CABs score and individual CABs questions. Additionally, since the CITIES Project only surveyed Ohioans, generalizability of this study can be improved by sampling different geographic regions. Furthermore, longitudinal studies can better examine changes in CABs and cancer knowledge before and after experiencing cancer within the family.

## Author Contributions

**Conceptualization:** Li Lin, Xiaochen Zhang, Mengda Yu, Brittany Bernardo, Toyin Adeyanju, Electra D. Paskett.

**Data curation:** Xiaochen Zhang, Mengda Yu, Brittany Bernardo.

**Formal analysis:** Xiaochen Zhang, Mengda Yu.

**Funding acquisition:** Li Lin, Xiaochen Zhang, Brittany Bernardo, Electra D. Paskett.

**Investigation:** Li Lin, Electra D. Paskett.

**Methodology:** Li Lin, Xiaochen Zhang, Mengda Yu, Electra D. Paskett.

**Project administration:** Brittany Bernardo, Toyin Adeyanju, Electra D. Paskett.

**Resources:** Xiaochen Zhang, Brittany Bernardo, Toyin Adeyanju, Electra D. Paskett.

**Supervision:** Xiaochen Zhang, Brittany Bernardo, Toyin Adeyanju, Electra D. Paskett.

**Validation:** Li Lin, Xiaochen Zhang, Electra D. Paskett.

**Visualization:** Li Lin, Brittany Bernardo.

**Writing – original draft:** Li Lin.

**Writing – review & editing:** Li Lin, Xiaochen Zhang, Mengda Yu, Brittany Bernardo, Toyin Adeyanju, Electra D. Paskett.

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
