## [Decision Letter · Decision Letter 0]

22 Nov 2022

PONE-D-22-03335The Relationship Between Family History of Cancer and Cancer Attitudes & Beliefs Within the Community Initiative Towards Improving Equity and Health Status (CITIES) CohortPLOS ONE

Dear Dr. Lin 

Thank you for submitting your manuscript to PLOS ONE. After careful consideration, we feel that it has merit but does not fully meet PLOS ONE’s publication criteria as it currently stands. Therefore, we invite you to submit a revised version of the manuscript that addresses the points raised during the review process.

Please remove Wald from 95 % CI in tables titlesReferences in the conclusion sec may be moved to the discussion part with focus on conclusion only.203 line (Backward Model Selection) remove it and add it either in the footnote or textreplace N with small n (N=603) I all the placescorrect the spacing of all tables’ titlesthere is difference and bias introduced while using multiple data collection methods like survey through phone calls, in-person interviews, and web

Kind regards,

Uzma Shamsi

Academic Editor

PLOS ONE

Journal Requirements:

“This project was supported by The Ohio State University Comprehensive Cancer Center using Pelotonia funds, a supplement to the NCI grant (P30CA016058), The Ohio State University Center for Clinical and Translational Science CTSA grant UL1TR002733, and supported by the Recruitment, Intervention and Survey Shared Resource (RISSR) at The Ohio State University Comprehensive Cancer Center (P30CA016058). This work was supported by the Samuel J. Roessler Memorial Scholarship through The Ohio State University College of Medicine’s Medical Student Research Program Scholarship (To LL) and the National Cancer Institute (F99CA25374501 to XZ). The funders had no role in study design, data collection and analysis, decision to publish, or reparation of the manuscript.”

Reviewers' comments:

Reviewer's Responses to Questions

**Comments to the Author**

1. Is the manuscript technically sound, and do the data support the conclusions?

Reviewer #1: Yes

2. Has the statistical analysis been performed appropriately and rigorously? 

Reviewer #1: Yes

3. Have the authors made all data underlying the findings in their manuscript fully available?

Reviewer #1: Yes

4. Is the manuscript presented in an intelligible fashion and written in standard English?

Reviewer #1: Yes

5. Review Comments to the Author

Reviewer #1: The article highlights an important association between family history of cancer and cancer attitude and beliefs.

There are some aspects that authors may consider reviewing

Participants eligibility criteria is not defined well. Under outcome variables why prostrate screening is not mentioned.

I don't see description for sample size calculation

2nd line in the discussion section on " We hypothesized that participants with a FDR with cancer would have more negative attitudes and beliefs about cancer" contradicts with line # 95-96 in introduction where FDR with cancer has been cited to related with more conscious attitude towards health screening.

Discussion

Overall length of this section seems very limited. Authors can consider adding more literature to compare and contrast the findings.

6. PLOS authors have the option to publish the peer review history of their article (what does this mean?). If published, this will include your full peer review and any attached files.

Reviewer #1: **Yes: **Nousheen Akber Pradhan

---

## [Author Response · Author response to Decision Letter 0]

12 Mar 2023

Thank you for your thoughtful review, comments, and suggestions. We believe we have now addressed these concerns and have detailed them below. 

Academic Editor comments: 

• Please remove Wald from 95 % CI in tables titles

RESPONSE: Wald has been removed from all table titles.

• References in the conclusion section may be moved to the discussion part with focus on conclusion only.

RESPONSE: References in the conclusion section have been removed and now are addressed in the discussion section.

• 203 line (Backward Model Selection) remove it and add it either in the footnote or text

RESPONSE: Thank you, (Backward Model Selection) was removed and added to the footnote for all relevant tables.

• replace N with small n (N=603) in all the places

RESPONSE: All N’s have been replaced with small n’s. 

• correct the spacing of all table titles

RESPONSE: Thank you, we have changed the spacing to single spacing for all table titles, if there are other concerns, we are more than happy to edit with more clarification from the reviewer. 

• there is difference and bias introduced while using multiple data collection methods like survey through phone calls, in-person interviews, and web

RESPONSE: Thank you, we agree with this comment. The strengths and limitations section has been updated to reflect this. 

Additional requirements:

RESPONSE: Thank you. This has been addressed.

“This project was supported by The Ohio State University Comprehensive Cancer Center using Pelotonia funds, and a supplement to the NCI grant (P30CA016058), The Ohio State University Center for Clinical and Translational Science CTSA grant UL1TR002733, and supported by the Recruitment, Intervention and Survey Shared Resource (RISSR) at The Ohio State University Comprehensive Cancer Center (P30CA016058). This work was supported by the Samuel J. Roessler Memorial Scholarship through The Ohio State University College of Medicine’s Medical Student Research Program Scholarship (To LL) and the National Cancer Institute (F99CA25374501 to XZ). The funders had no role in study design, data collection and analysis, decision to publish, or reparation of the manuscript.”

RESPONSE: the Financial Statement has now been updated and is included in the cover letter.

RESPONSE: Thank you. This has been addressed.

RESPONSE: Thank you. This has been addressed.

Reviewers' comments:

Reviewer #1: The article highlights an important association between family history of cancer and cancer attitude and beliefs. There are some aspects that authors may consider reviewing

Participants eligibility criteria is not defined well. 

RESPONSE: Data for the current study was collected as part of the CITIES project. The overall goal of the CITIES project was to define and describe the catchment area of the Ohio State University Comprehensive Cancer Center (OSUCCC). All Ohio residents between the ages of 21 to 74 were eligible for this study. The methods section has been updated to correct the age range. A variety of recruitment methods were used to gather a diverse sample. We briefly summarized the sampling methods in the manuscript. More detailed methods and sampling strategies have been described in the original CITIES manuscript and is cited in the methods:

Paskett ED, Young GS, Bernardo BM, Washington C, DeGraffinreid CR, Fisher JL, et al. The CITIES Project: Understanding the Health of Underrepresented Populations in Ohio. Cancer Epidemiol Biomarkers Prev. 2019;28(3):442-54.

Under outcome variables why prostate screening is not mentioned.

RESPONSE: Thank you for the thoughtful comment. Knowledge about prostate cancer screening was not included as a question in the CITIES survey; therefore, it could not be evaluated as an outcome in our study. Notably, prostate cancer screening guidelines are highly variable. Screening recommendations depend on an individual’s life expectancy and shared decision-making between patient and their provider. 

I don't see description for sample size calculation

RESPONSE: A sample size calculation was not performed for this study. We considered if we would have an adequate sample size for analysis by considering the minimum counts needed per predictor. The maximum number of predictors used in the models was 9 variables, and with the minimum of 10 counts per predictor, the study is well over the sample of n=90 needed for analysis. 

2nd line in the discussion section on " We hypothesized that participants with a FDR with cancer would have more negative attitudes and beliefs about cancer" contradicts with line # 95-96 in introduction where FDR with cancer has been cited to related with more conscious attitude towards health screening.

RESPONSE: The studies cited report a positive relationship between having a FDR with cancer and increased cancer screening behavior. It is thought that cancer attitudes and beliefs (CABs) may play a role in this association, but there are no studies to directly support this. Our study specifically examines the relationship between FDR and cancer attitudes and beliefs. 

Discussion

Overall length of this section seems very limited. Authors can consider adding more literature to compare and contrast the findings.

RESPONSE: Thank you for your careful review and suggestion. The discussion section is updated with additional literature.

---

## [Editor Report · Decision Letter 1]

11 Apr 2023

PONE-D-22-03335R1The relationship between family history of cancer and cancer attitudes & beliefs within the Community Initiative Towards Improving Equity and Health Status (CITIES) cohortPLOS ONE

Dear Dr. Lin,

Thank you for submitting your manuscript to PLOS ONE. After careful consideration, we feel that it has merit but does not fully meet PLOS ONE’s publication criteria as it currently stands. Therefore, we invite you to submit a revised version of the manuscript that addresses the points raised during the review process.

1. The justification provided is not entirely correct. While it is true that having a minimum count of 10 per predictor is a general rule of thumb, it is not the only consideration when determining sample size for a study. Other factors that should be taken into account include the effect size you want to detect, the desired level of statistical power, and the alpha level (level of significance) you plan to use. Without a formal sample size calculation, it is difficult to determine if the sample size is truly adequate for the study's research questions and hypotheses. Therefore, while the study might have enough data to conduct some form of analysis, the lack of a formal sample size calculation could be considered a limitation of the study.

2. Tabs. 3 & 4 results are not mentioned in the abstract and are not coherent with the main study objectives.

3. Need to improve the main focus of the manuscript with coherence in the objectives and results.

Please submit your revised manuscript by May 26 2023 11:59PM.  If you will need more time than this to complete your revisions, please reply to this message or contact the journal office at plosone@plos.org. Please include the following items when submitting your revised manuscript:A rebuttal letter that responds to each point raised by the academic editor and reviewer(s). You should upload this letter as a separate file labeled 'Response to Reviewers'.A marked-up copy of your manuscript that highlights changes made to the original version. You should upload this as a separate file labeled 'Revised Manuscript with Track Changes'.An unmarked version of your revised paper without tracked changes. You should upload this as a separate file labeled 'Manuscript'.If applicable, we recommend that you deposit your laboratory protocols in protocols.io to enhance the reproducibility of your results. Protocols.io assigns your protocol its own identifier (DOI) so that it can be cited independently in the future. For instructions see: https://journals.plos.org/plosone/s/submission-guidelines#loc-laboratory-protocols. Additionally, PLOS ONE offers an option for publishing peer-reviewed Lab Protocol articles, which describe protocols hosted on protocols.io. Read more information on sharing protocols at https://plos.org/protocols?utm_medium=editorial-email&utm_source=authorletters&utm_campaign=protocols.

We look forward to receiving your revised manuscript.

Kind regards,

Uzma Shamsi

Academic Editor

PLOS ONE
---

## [Author Response · Author response to Decision Letter 1]

25 May 2023

Thank you for your thoughtful review, comments, and suggestions. We believe we have now addressed these concerns and have detailed them below. 

1. The justification provided is not entirely correct. While it is true that having a minimum count of 10 per predictor is a general rule of thumb, it is not the only consideration when determining sample size for a study. Other factors that should be taken into account include the effect size you want to detect, the desired level of statistical power, and the alpha level (level of significance) you plan to use. Without a formal sample size calculation, it is difficult to determine if the sample size is truly adequate for the study's research questions and hypotheses. Therefore, while the study might have enough data to conduct some form of analysis, the lack of a formal sample size calculation could be considered a limitation of the study.

RESPONSE: We acknowledge that we did not have a sample size calculation for this study. However, the goal of the Community Initiative Towards Improving Equity and Health Status (CITIES) project was to develop core survey items and implement population surveys in the catchment area of the OSU Comprehensive Cancer Center. Since this was a one-time cross-sectional survey conducted between 2017 and 2018, and we are not quantifying any intervention effects or conducting comparisons to other populations, we do not think a post hoc sample size calculation is required. The current study is to describe cancer attitudes and beliefs among the study population within the OSUCCC catchment area. We included that the findings from this study is not generalizable to other populations as a limitation. We also described the differences of the study sample characteristics within the general Ohio residents. Therefore, we think we have addressed the limitation of the study design. 

2. Tabs. 3 & 4 results are not mentioned in the abstract and are not coherent with the main study objectives.

RESPONSE: The manuscript has been revised to include the importance of tables 3 & 4. 

3. Need to improve the main focus of the manuscript with coherence in the objectives and results.

RESPONSE: The manuscript has been revised to address coherence.

---

## [Editor Report · Decision Letter 2]

12 Jun 2023

The relationship between family history of cancer and cancer attitudes & beliefs within the Community Initiative Towards Improving Equity and Health Status (CITIES) cohort

PONE-D-22-03335R2

Dear Dr. Lin,

We’re pleased to inform you that your manuscript has been judged scientifically suitable for publication and will be formally accepted for publication once it meets all outstanding technical requirements.

Kind regards,

Uzma Shamsi

Academic Editor

PLOS ONE
---

## [Editor Report · Acceptance letter]

19 Jun 2023

PONE-D-22-03335R2 

The relationship between family history of cancer and cancer attitudes & beliefs within the Community Initiative Towards Improving Equity and Health Status (CITIES) cohort 

Dear Dr. Lin:

I'm pleased to inform you that your manuscript has been deemed suitable for publication in PLOS ONE. Congratulations! Your manuscript is now with our production department. 

Kind regards, 

on behalf of

Dr. Uzma Shamsi 

Academic Editor

PLOS ONE